# No Free Lunch GPT/LLM Limitations

**Serge Berger**
AIDIL
Mill Creek, WA 98012, USA
csergeb@aidillic.com

## Abstract

Recently, AI models (LLM/GPT) have demonstrated their reasoning capabilities in rigorous settings and their utility as research assistants. However, in this paper, we reveal a limitation of this approach in generating rigorous proofs (including with the help of RLHF). This limitation applies to GPT/LLM-agentic and hierarchical models, as well as to a subset of EBM models.

## 1 Introduction

In this paper, we demonstrate that reasoning/proving inference issues in modern AI models are inherent logical consequences of their architectures (i.e., GPT/LLM). We formalize the problem of generating valid inference into a domain of finite graphs, $\mathcal{G}(\omega)$, and prove the following general theorem: *For almost all proofs, any learning algorithm of inference using randomization in $\mathcal{G}(\omega)$ is (asymptotically) almost surely literal learning.*[1]

**"Literal learning"** (as a method of training/inference) stands for one that is

- either valid and memorizes the inferences from a corpus of training/synthetic data (i.e., the memorized or formalized proofs), or

- vacuous, i.e., $\forall x\, [P(x) \implies Q(x)]$ where $P(x)$ is false for every $x$, or

- creates a random inference from a false assumption (hallucination), or

- otherwise invalid.

In other words, the theorem above states that an architecture based on GPT/LLM almost surely fails to produce a valid proof for a sufficiently complex statement.

On the one hand, the GPT architecture for LLMs has demonstrated significant progress in the generative manifestation of summarizations, chat, and material representations. On the other hand, the architecture displayed multiple adverse effects, such as hallucinations, obduracy, falsehoods, degraded generalization, supercilious attitude, and performance degradation (e.g., (Yadlowsky et al., 2023)). In rigorous contexts (where one requires a consistent mathematical reasoning or a formal proof), the results are consistently discomforting (Chen et al. (2023), Hagendorff et al. (2022), and Dziri & et. al. (2023)).

Recently, several authors have pointed out various limitations (cf., e.g., (Liu et al., 2023), (Mikhaylovskiy & Churilov, 2023), and (Asher et al., 2023)). Nonetheless, possible remedies have been suggested ((Sel et al., 2023), (L. et al., 2023), and (Z. et al., 2022)).
In this paper, we show that these issues are an inherent *logical* consequence of the GPT architecture. As a result, multiple phenomena of transformer inference limitations can be explained from a purely logical view; in particular, some results of (Dziri et al., 2023) can be obtained that way. It is demonstrated that some limitations addressed in the paper (e.g., the issue of increasingly large parallelism requirements) can be alleviated by modifying the type of attention.

---

[1]Note that we say "almost surely" we mean "asymptotically almost surely" since only the latter makes sense. The reader can find an instructive explanation of the prime number theorem later in the paper.

## 2 MOTIVATION/PRELIMINARIES

To demonstrate the model that proves the inherent reasoning limitation phenomena, we need to formalize logically the mechanism employed by GPT/LLM to predict the next token in a sequence. It turns out that randomization used by *GPT* architecture in the sampling procedure (e.g., "logits temperature scaling") is the main reason. However, it isn't easy to devise an alternative when dealing with training on a large text corpus. Note that the architecture becomes too "predictable/plain" if we choose the most probable pattern in the list of candidates for the next token. There, we would need a specific randomization to become "innovative".

Notations and terms can be found in Appendix E . Literal learning is the main concept. To develop intuition for the exact nature of the claims, we begin with instructive informal considerations. For that reason, the proofs are marked "informal". Otherwise, in a rigorous setting, the tag is "formal".

### 2.1 DEFINING THE CONTEXT

We can assume that one can enumerate all the inferences using $[n]$, since there is only a countable number of (finite) proofs on a countable number of entities. For a graph in $\mathcal{G}(\omega)$, define property $\mathcal{A} := \{\exists \text{ nodes } e_1, \ldots e_j \text{ forming chain } (e_0 \to e_1 \ldots e_i \ldots e_j \to e_m) \text{ for inference } e_0 \to e_m)\}$. This definition is well-formed since $\mathcal{A}$ is expressed as a first-order formula in the first-order logic theory for $\mathcal{G}(\omega)$), and the axiom of foundation[2]. For chains above, we need to verify that these are first-order expressions. A suitable framework for this is that of least fixed point extension (cf. (Grohe, 2017)). If the " $\sim$ " is a connectivity relation, then a chain $C(e_0, e_t)$ where $e_0$ and $e_t$ represent a proof starting node $e_0$ and terminal node $e_t$ respectively, can be expressed as follows:

$$C(e_0, e_t) \leftarrow ((e_0 = e_t) \vee \exists e_i (C(e_0, e_i) \wedge e_i \sim e_t)) \tag{1}$$

Then, by Lemma C.1, we have two possibilities for probability: $p = \lim_{n \to \infty} \mathbb{P}(G_n(\omega) \in \mathcal{A}) = 0$ or $p$ is equal to 1 ($G_n(\omega) = \{G : |G| = n \text{ and } G \text{ is a graph}\}$). If it is zero, then, almost surely, no valid proof can be found within the context. Therefore, $\lim_{n \to \infty} \mathbb{P}(G_n(\omega) \in \mathcal{A}) = 1$. By Lemma 0 in C.3, it follows that $\mathcal{G}(\omega) \models \mathcal{A}$. However, it also means that our inference follows a literal graph representation from the original (i.e., from the given training set). Similar considerations apply to a novel versus a non-novel context. Thus, we must assume $p \neq 1$. In this case, we create a hierarchy in $\mathcal{G}(\omega)$ as follows.

Consider chain $(e_0 \xrightarrow{\psi_1} e_1 \ldots e_i \ldots e_j \xrightarrow{\psi_k} e_m)$ and formula $\psi := \psi_1 \wedge \cdots \wedge \psi_k$. Clearly, $\psi$ is true in $\mathcal{G}(\omega)$ for any inference of $e_m$. But that means that we again have a "literal" learning. Otherwise, since $p$ is not 1, we will have a "fault" for sufficiently large $n$.

In (Blass et al., 1998), the authors proved a version of the zero-one law for binary sequences, as well as a decision problem within this context. Our formal proof is a generalization to a class of algorithms in which logical inference admits a standard finite graph representation. We just (semi-formally) proved the following:

**Theorem 2.1.** *For almost all proofs, any valid learning algorithm of inference, using randomization in $\mathcal{G}(\omega)$, is almost surely literal learning.* ∎

The complete formal proof is in D . Previously, we established a natural model for inference and pointed out the limitations associated with it. We will see below that, in other words, the algorithm almost surely (a.s.) fails unless its inference is a literal learning.

**Theorem 2.2.** *(reformulation of Theorem 2.1) Given the graph model of inference for machine learning, $G(\omega)$, the only algorithm based on randomization that also maintains veracity of inference is almost surely literal learning. In other words, for a sufficiently long proof, any algorithm that randomly deviates from the training data (a.s.) will fail to generate a valid proof.* ∎

---

[2]In a second-order logic, one can quantify over sets of domain elements; in the first-order logic, one can quantify over elements only.

Note that this means the post-training techniques (such as Deepseek MOE, MLA, FP8, or MTP) are no remedy, as well as reinforcement learning or HFRL on a distilled student model.

**Corollary 2.3.** *Within a rigorous inference context, almost surely, no randomization of the prediction scheme of proof patterns in $\mathcal{G}(\omega)$ can discover new (unknown) non-trivial valid statements of medium or high complexity.*

*Proof.* This can be easily explained: since any degradation is inherited in the foundational graph, the subsequent inferences on the trained data tend to deviate from already shortened erroneous paths, thus multiplying the faults. ∎

**Theorem 2.4.** *(Generalization of the main result) Any algorithm of learning enforcing veracity, admitting a 0-1 domain cast as random graphs (i.e., admitting 0-1 law), is almost surely memorization.*

*Proof.* This is the context where proof of Theorem 2.1 is fully applicable. ∎

Within the view adopted herein, there is an interesting example of a decidable theory that admits a 0-1 random graph domain. Yet, its classifier comparison is not expressible in its first-order logic. Therefore, it is an example of a.s. learning algorithm with randomization, which is ultimately decidable but vacuous in the sense it does not support any notion of expressible first-order classifier comparison. Thus, there is no feasible notion of fairness for classification tasks. The classifier is not based on a measure of standard probability space (cf. appendix F.3).

Incidentally, the paper (Trinh et al., 2024) depicts good results on solving geometry problems of Olympiad level. First, we have to note that, because the first-order theory of Euclidean geometry is elementary in the logical sense (decidable), the task is achievable by a universal algorithm since we can work in the decidable first-order theory of $\mathbb{R}$. For (analytic) geometric problems involving constructions, such an algorithm in its explicit form has been known since Descartes.

Since the output is in natural language (rather than code for an automated prover, unlike the approach of (Zheng et al., 2022)), it isn't easy to assess the solution's performance. Because this transformer is trained on synthetic data and proofs are relatively short by nature of the problems involved, and because of our main result, the solution is likely not to exceed a threshold of vacuous/literal learning overall. A more formalized approach is presented in (Krueger et al., 2021).

In (Nezhurina et al., 2024), there are more examples of basic reasoning breakdowns for foundational industrial models.

In (Nguen & Sarah, 2022), the authors describe multiple patterns of software development that reflect erroneous or sub-optimal code generated by Copilot. This leads to elevated code churn and downward pressure on code quality on $GitHub$. Another survey, (Kabir et al., 2024), shows that coding questions generate up to 50% of errors. A similar study is conducted in (Macmillan-Scott & Musolesi, 2024).

## 2.2 DISCUSSION

The results above easily explain the phenomenon of "hallucinations" and brittleness of the GPT models in a rigorous context. It also means that LLMs is unlikely to discover any new mathematical result of sufficient strength unless the GPT/LLM model takes the role of an assistant.

In (Gendron et al., 2023) shows that the construction of the baseline dataset for rigorous learning should be a formal exercise. Consider the task of equation completion in which one has to predict a missing symbol. Since this aligns perfectly with the main premise of LLM based on transformers, one can expect a success rate for this task to be quite high. As is shown in the paper, this is not the case. An associated (and well-known) phenomenon, where a plateau of performance is followed by subsequent exponential degradation, manifests in the same way as in the generic sequence case. Similarly, a few authors summarize a few problems in the answers to contemporary systems regarding low P/R with respect to citation usage from the underlying sources. These experimental results are not for the rigorous context.

It is well-known that any mathematical problem of significance requires one or multiple critical insights that are just not to be found. These are not combinations of known results (or tactics), but relatively completely new, albeit inevitable, ideas.

For instance, to address some long-standing problems, new fields of mathematics had to be created, thereby representing a new body of knowledge. Thus, generalizing the LLM solution for these targets is a task of yet another level of complexity for which the method is not suited. Moreover, as we show below, it is (almost surely) guaranteed to fail. The inevitable conclusion is that the apt inference model has to be more deferential to logic and understanding.

Our main theorem is yet another piece of evidence that, "under normal conditions," generating a proof is driven by its logical complexity. In practical terms, this means that machine learning or AI algorithms are insufficient for generating complex (mathematical) proof since they are only concerned with the statistical properties of data. By the same token, they may be used to create instrumental interactive assistants for working mathematicians and computer scientists, since the assistants are likely to extract essential patterns in proofs and code.

However, the problem may be even more complex. We have the following observation.

**Proposition 2.5.** *There is a formula $\theta$ such that $\phi_n = \phi(n)$ is valid in in the first-order logic theory $\Psi$ for every n, but $\theta = (\forall n)\, \phi(n)$ is undecidable.*

*Proof.* Without loss of generality, we can assume that $\theta \in \Psi$ and $\phi_n$ encode the statements of a first-logic theory $T$ that includes arithmetic. More specifically, $\phi_n$ encodes the statement that no contradiction is introduced into $T$ by a proof of length less than or equal to $n$. If $T$ is consistent, then every $\phi_n$ is provable in $T$, and $\theta$ implies consistency of $T$. However, by Gödel's second incompleteness theorem, that cannot be proven in $T$.

A more intuitive idea for the shorter proof of the proposition is to assign a proof $\phi_n$ its complexity and order these complexities in ascending order. The complexity for $\theta$ will exceed all of them. That means $\theta$ cannot be proved in $T$. ∎

Thus, any generator/prover must have an overarching strategy to resolve the undecidability issue and generate the proof for $\theta$. There is a good analogy to other mathematical situations. For instance, the insolvability of the quintic is proved by the observation that a possible solution in radicals, in general, will require an infinite number of nested radicals.

We complete the discussion with an instructive analogy. One of the useful formulations of the prime number theorem states that the probability of a number, randomly picked from the interval $[1, n]$, being prime is approximately $1/ln(n)$. In other words, for sufficiently large $n$ it is almost surely zero. It is easy to see that these sufficiently large numbers are quite large to yield probabilities virtually guaranteeing that the number is composite. It is not unlike our main result, given the complexity of the proof, but the analogy stops at the term "sufficiently complex proof".

The complexity, in plain terms, can be from medium to hard, and it is virtually guaranteed that the novel result admitting only hard proof is not going to be correctly generated and proved. The reader should be careful to have a clear understanding of the reformulation of our result in terms of insolvability, a sufficiently complex problem, almost surely, since there have recently been several claims that AI assistants greatly help solve novel, challenging problems. "Assisting" is a complex two-way process that can be described as a form of $RLHF$. Therefore, any analysis of process performance must be adjusted accordingly.

Specifically, the analogy with the prime number theorem survives if we consider a few trial events looking for a series with length $k$ not containing primes at all. The probability is a polynomial over $k$. In our case, it is exponential. It is essential to have a concrete number for the length of proof when success becomes virtually impossible. One proper representation is Lean 4 (or Coq) code generation for the proof. This is useful for verification; however, it creates a noticeable complexity overhead.

## 2.3 LIMITATIONS AND CLARIFICATIONS

At this point, a couple of natural questions may arise.

**Q**: Given the main result of the paper, how does one address the evidence that some models have produced valid proofs not seen in the training set?

**A**: The main result is statistical evidence. It does not guarantee that, given a statement, the inference will always be memorization. It only states that, if valid, this will happen almost surely for a sufficiently complex problem. From a practical standpoint, it means that, given a statement of sufficient complexity, the failure will occur with high probability (tending to 1 as the proof's complexity increases). The same conclusion applies to all known improvements, such as greedy decoding, COT prompting, retrieval-augmented generation, and context synthetic data, since LLM/GPT is the base inference algorithm.

A standalone case of proof assistants (and the adjacent technique of HLRF verification) is simply out of scope, as it involves deterministic inference. Clearly, it is theoretically possible to create a model that generates verified Coq or Lean code. The main result then shows that this model will not be based on LLM/GPT; however, given a rich library of proofs, we can expect limited success in generating valid proofs within the scope of a suitable library, such as contemporary code generation for popular programming languages.

**Q**: Why cannot the same argumentation be applied in a general case? After all, can any chain of thought or deductive reasoning be represented by a graph chain?

**A**: There are two keys in the reasoning. First, GPT needs randomization. It is simply not the case in a general mathematical setting. Similarly, the stochastic algorithm would produce a series of errors – there is no intermediate verification that prevents the construction of false implications – the veracity of the chain is only a property of the first-order logic model, which the algorithm disregards.

Logically, proof is a path in $G_L$(chain) of "true" connections between nodes represented by factual statements in a suitable (holistic) enumeration. Once we encounter a false statement, it is no longer proof; at best, it is a vacuous truth, which is no longer a valid proof. That may be a source of hallucinations. One may consider proof by contradiction; however, that case can be easily reduced to the above situation.

More specifically, any chain (or path) in $\mathcal{G}(\omega)$ must be of the form $e_0 \to \ldots \xrightarrow{\phi} e_t$ where $\phi$ is a valid implication. For the proof by contradiction, we may recall that any implication $a \to b$ is $\neg a \vee b$. Taking $a \to b$, assume that $b$ is not true. Then $\neg b$ is true. There must be a chain $e_0 = \neg b \to e_t$ where $e_t$ is a contradiction of the form $a \wedge \neg a$. Then $e_0 \to e_t$ is only valid when $\neg e_0 \vee e_t$ is true, which means $\neg e_0$ is true. Thus, we have reduced the proof to the general case.

The proposition 2.5 provides another example in which stochastic inference will not yield the correct result unless the overarching strategy is known.

**Q**: Why is the accumulating error lemma (A.1) is not sufficient for proving the main result?

**A**: Practical considerations. Even though the lemma explains many phenomena of erroneous inferences, it is not sufficient to prove the main result. The reason is that the rate of exponential decay can be extremely low, and the manifestation of having an error for sufficiently complex proof may not be computationally tractable.

Combined with a logical argument, the main result provides a tractable estimate and a practical method for uncovering the computational limitations of transformer architectures in reasoning.

**Q**: The error accumulation model should account for correlated reasoning steps and self-correction mechanisms. Also, modern architectures are more complex than the first-order logic view.

**A**: Indeed, these mechanisms include multiple heuristics, such as second-of-multipass, verifier-in-the-loop, model critique, and other refinements. Yet none of these refute the premises of the accumulation Lemma for a simple reason. These mechanisms are error-prone, and with a large enough window (since it is finite), the premises of the accumulation Lemma remain true. With an appropriate level of abstraction, the first-order model captures the correctness of higher-logic constructs and structures. Therefore, the adopted model suffices.

## 3   CONCLUSION

One noticeable phenomenon in efforts to improve the reasoning capabilities of LLM architectures is the variation of sampling and the use of a few adopted performance benchmarks (e.g., (Muenninghoff et al., 2025)). A conceptual survey on (inherent) reasoning limitations can be found in (Russinovich et al., 2025). They correctly state that hallucinations, jailbreaking, and falsehoods are inherent limitations of the stochastic LLM architecture (autoregression and stochastic gradient descent). We can only add that there are many mixed phenomena due to the loss of model context, which lead to unavoidable faults in sufficiently long proofs or those of sufficiently high complexity.

It is error-prone to measure performance with just a few well-known benchmarks (e.g., MATH, MiniF2F, AIME, and Similar). Flavors of the problems and their solutions can be found even in small samples (or just in reformulations with trivial modifications). Thus, the model may easily fall into vacuous/literal learning. With the advancement of synthetic data generation by modern formal assistants, it is unrealistic to be confident that a model has not simply memorized its formalizations and proofs (potentially modified by re-aggregation).

For this reason, to avoid memorization, we have to use advanced (novel) benchmarks with newly formulated problems that incorporate original ideas. It is inconvenient because the authors of new approaches may lose a baseline; however, it is the only way to ensure tangible progress.

Such efforts, however, are already underway ((Glazer et al., 2024)). Unlike other benchmarks, which reach saturation with an unsolved rate below 50%, FrontierMath maintains an unsolved rate above 98% (2024-5).

Recently (2026), an announcement stated that an LLM-based tool (Aristotle by Harmonic) solved the Erdős problem 124, a conjecture that has remained open for nearly 30 years since it was first posed in a 1995 paper titled "Complete sequences of sets of integer powers" in the journal Acta Arithmetica.

Here, the reader has to take into account the number of problems posed by Erdos (944; more than 300 have been solved so far). Another example is the problem no. 728 ((Sothanaphan, 2026)). The Lean-based proof is used, so it seems to be a good test case for our heuristic estimate of the critical threshold for valid inference. However, Aristotle's approach is based on popular Lean libraries created by a human community.

Even the natural-language proof was reverse-engineered by a human agent from Lean after crucial interventions by human experts. Thus, our estimate is only applicable in the form $(threshold = \alpha + \beta)$ where $\alpha = 0$ is the number of lines in libraries inference depends on is nullified, plus $\beta$, which is our estimate ($beta$ for the problem no. 728 is less than 1.5K Lean lines « 10K threshold in our heuristic study).

Moreover, upon closer inspection, these problems admit a relatively simple solution, which is obscured by extraneous factors beyond the scope of this paper's central theme (e.g., the level of researchers' interest in a particular unsolved problem). Therefore, it correlates with our fundamental prediction that an LLM-based approach would almost surely fail for sufficiently complex problems.

For instance, the success of AI assistants and LLM-based tools in solving problems from mathematical olympiads is easily explained by the domination factors like "closedness" of the elementary (formal) system behind the solutions, a priori fact of existence of a relevant solution reproducible in a limited time, and most importantly, the elementary character (elementary classes, decidability and completeness) of the domain's first order logic models.

And the most critical lesson: LLMs assistants demonstrated their value in collaboration with an expert. That has been proven in many areas before, e.g., in Computer Science. Chess is an obvious example. Demonstrating a better play than humans (mostly, by achieving a deeper ply horizon in alpha-beta search and due to a human-generated highly specialized cost function), none of the methods even try to answer the most compelling question: "Is the initial position a draw?" The answer is complex due to the extremely high-dimensional space of the finite game first-order logic model.

We can only recommend going to more focused contexts where one formulates problems in specific mathematical domains, e.g., the Scottish Book in topology, the Kourovka Notebook in algebra, open problems in algebraic number theory, and geometry. It appears that when the process becomes user-friendly, most professionals will use AI frameworks to advance their research.

Today, they are error-prone and slow, and progress in applying the tools is slow. Yet, our result shows that there is no free lunch. There will always be an issue with generating trustworthy inference for sufficiently complex problems. From this standpoint, it is paramount to continue research in this direction and to deepen understanding the rich nature of AI's limitations in reasoning.

Practitioners (architects) are realizing that there is a separation of concerns. Humans create innovative and novel architectures, and AI tools handle the compliance and rules of good design in complex architectures.

Mathematicians are already in that realm. However, it is not just a stage. It is a ceiling on hyperscalers' approach until automated proving reaches its own maturity.

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

APPENDIX

## A MAIN RESULT

In this section, we assume a natural representation of a proof by a path from a node $e_0$ to the node $e_t$ in the graph $G(\omega)$ in appendix C.1, which, in a suitable enumeration, corresponds to a premise/target statement $e_0, \ e_t$ accordingly.

**Lemma A.1.** *(Accumulating Errors Lemma). Assuming independence of faults in $\mathcal{G}(\omega)$ representing proofs, and a nonzero probability of fault at each step, the probability of a fault-free proof tends to zero exponentially over its length.*

*Proof.* We can guarantee zero probability of fault only if the proof is holistically embedded in the training resources and then followed by the inference algorithm. Such greedy inference represents a trivial case, a literal learning.

Moreover, we can assume that faults are independent, since the semantics of a formal inference are outside the scope. [3] It has no notion of non-statistical meaning. Let the labeled graph $\mathcal{G}(\omega)$ represent proofs (chains) of enumerated statements (nodes) where each label is a probability $p_i$ that the chain ending with the corresponding node contains an error. Then we have:

$$\mathbb{P}(no\ fault\ proof) \leq exp(-\mathbb{E}(number\ of\ faults)) \tag{2}$$

There are multiple ways to prove this using Jensen and Markov inequalities. The elementary proof using just the independence of faults is as follows. Let $I_i$ be an indicator of a fault for event indexed $i$ and the number of faults $X = \sum_{i=1}^{n} I_i$. Then we have:

$$\mathbb{P}(no\ fault\ proof) = \mathbb{P}(X = 0) =$$
$$\mathbb{P}(I_1 = 0, \ldots, I_n = 0) = \prod_{i=1}^{n}(1 - p_i). \tag{3}$$

The standard inequality $1 - p \leq e^{-p}$, applied to each factor, leads to:

$$\mathbb{P}(no\ fault\ proof) = \prod_{i=1}^{n}(1 - p_i) \leq \prod_{i=1}^{n} e^{-p_i} =$$
$$e^{-\sum_{i=1}^{n} p_i} = e^{-\mathbb{E}[X]}. \tag{4}$$

By assumptions of non-zero fault probability, since the right side of the equation 2 tends to zero, we have:

$$\lim_{n \to \infty} \mathbb{P}(no\ fault\ proof\ of\ length\ n) = 0. \tag{5}$$

This completes the proof. ∎

Note that this is a generalization of Lemma D.2 in (Dziri et al., 2023) where the respective result is proven under restrictive assumptions. Moreover, the main results of the paper, propositions P.1 and P.2, follow from our main theorem, D, under the assumption of the main scenario, GPT/LLM. By the same token, the Lemma applies to any learning algorithm considered in this paper (E).

*Example* 1. Since GPT has no semantic notion on the entities involved, we can assume the Lemma A.1 is fully applied to the GPT rigorous context for the following reason: given a sufficient length of the proof (e.g, the proof of high enough complexity, randomization will invariably arrive to a fragment of the evidence that does not pay attention to another fragment (cf. attention window length). In other words, no overarching strategy will be possible. Since our man result is formulated for an "almost surely" context, the assumption of sufficiently high complexity of a proof is satisfied.

**Corollary A.2.** *Assume an algorithm admitting model $\mathcal{G}(\omega)$ for inference and using randomization. Then the rate of (correctness) decay is exponential over the proof length.*

---

[3]GPT algorithm does not follow the syntax of the first-order theory – instead, it uses randomization and inferred statistics.

*Proof.* The proof is similar to a usual consideration for a set of independent events in a classic probability space generating a fault. The key observation is that once a fault in the chain of inference occurs, the chain is subsequently erroneous. This proves that the correctness rate decays exponentially over the length of the proof. ∎

**(Heuristic note)** In a few papers, this phenomenon has been shown experimentally. Moreover, it has a few incarnations. These include hallucinations (when there are no supporting references for an inference), erroneous statements (such as falsehoods, incorrect generalizations, or non sequiturs), and a general misalignment. Exponential decay is also noted in a few papers; our result (Corollary 2) shows that it holds for all non-trivial (complex enough) tasks, including performance degradation on synthetic data in an autophagous loop.

**Theorem A.3.** *(Inherent reasoning* GPT/LLM *Limitation).    Any algorithm of inference, using randomization on $\mathcal{G}(\omega)$, is almost surely literal learning.*

*Proof.* (Informal) We give two proofs of the statement. To develop a theoretical intuition, we start with the one below. The second one, rigorous and more instructive, is in D .

Note that we can assume that one can enumerate all the inferences using $[n]$, since there is only a countable number of (finite) proofs on a countable number of entities (statements). Without loss of generality, for that representation of entities, we can assume that node (vertices) $e_i$ implies $e_j$ only if they are connected; we do not need to impose any order on the nodes.

For a generative model, that would be the enumeration of proofs for a particular prompt, such as proving that $e_k$ implies $e_l$. Moreover, we can assume that a generic proof is an actual chain of thought, i.e., we have a finite sequence of distinct nodes connected via regular paths, with possible cycles that reflect the equivalence of the statements (we can always ignore the cycles). The underlying training graph for this is not necessarily connected; however, the model output must contain a path from the premise to the desired conclusion.

In appendix B, we define the first-order language of graphs used in an associated model, $\mathcal{G}(\omega)$.

Thus, the path (or "chain of thought") is just a sequence of tuples $(s_k \sim s_l)$ where the sign "$\sim$" represents adjacency for vertices $s_k$ and $s_l$, and there is a path.

$$e_s \sim e_1 \wedge e_1 \sim e_2 \wedge \cdots \wedge e_k \sim e_t \tag{6}$$

Now, there are two possibilities:

1. The path (6) exists in the training set (not a novel context).

2. The path (6) does not exist in the training set (a novel context).

Consider the first-order formula $\phi(.) = e_s \sim e_1 \wedge e_1 \sim e_2 \wedge \cdots \wedge e_k \sim e_t$. Then, again, we have two possibilities. By 0-1 law (C.1), we have:

$$\lim_{n \to \infty} \mathbb{P}(\mathcal{G}(\omega) \models \phi) = 0 \; or \; 1. \tag{7}$$

Thus, we have four possibilities:

1. The limit (7) is equal to zero and the path (6) does not exist in the training set.

2. The limit (7) is equal to zero and the path (6) exists in the training set.

3. The limit (7) is equal to one and the path (6) does not exist in the training set.

4. The limit (7) is equal to one and the path (6) exist in the training set.

For each of these, we also need to consider the cases where the model temperature, normalized to probability $p$, equals 0 or 1, or lies between 0 and 1. We can assume the following for these cases:

For the case **1**, it is nearly obvious that, within the context, no valid proof can be found almost for sure in the first place, either if we try literal learning, falling into a novel context, or varying the probability $p$ between zero and one - we apply Accumulating Errors Lemma since GPT is an accumulating errors algorithm. The latter manifests itself as a phenomenon of accumulating errors for sufficiently complex (lengthy) proofs.

The case **2** is more interesting. Despite having proof in the training set and a chance of literal learning, we use probability $p$ other than one. As a result, we are having the phenomenon of accumulating errors described above.

The case **3** is the most interesting – we are in a novel context – and may follow fragments of the proof, somehow creating the final proof as an assembly. Note that we chose $p$ equal to 1. It means that we are trying to assemble the required proof in pieces. The problem is equivalent to finding paths among potentially connected pieces. However, we can simply apply C.3 and note that since $p$ is equal to 1, we have:

$$\lim_{n \to \infty} \mathbb{P}(G(n) \models \phi) = 1 \Leftrightarrow \mathcal{G}(\omega) \models \phi. \tag{8}$$

Therefore, given the ever-growing complexity and length of proofs, we must follow these fragments literally, which means we include them in the training set. That is literal learning, or we have a contradiction with the assumption of this case.

The case **4** *is* literal learning, by definition.

The conclusion is that *almost for sure*, only literal learning has a chance of generating an error-free proof.

$\lim_{n \to \infty} \mathbb{P}(G(n) \in \mathcal{A}) = 0$ or it is equal to 1. If it is zero, then no valid proof can be found within the context in the first place. Therefore, $\lim_{n \to \infty} \mathbb{P}(G(n) \in \mathcal{A}) = 1$. By C.3, in the first-order theory for the language of graphs, it follows that $\mathcal{G}(\omega) \models \mathcal{A}$. For $p = 1$, it is possible. However, it also means that our inference follows a literal graph representation of the original (i.e., the given training set). Thus, $p \neq 1$. In this case, we create a hierarchy in $\mathcal{G}(\omega)$ as follows.

Consider chain $(e_0 \xrightarrow{\psi_1} e_1 \dots e_i \dots e_j \xrightarrow{\psi_k} e_m)$ and formula $\psi := \psi_1 \wedge \cdots \wedge \psi_k$. Clearly, $\psi$ is true in $G_\omega(p)$ for any inference of $e_m$. But that means that we again have a "literal" learning. Otherwise, since $p$ is not 1, we will have a "fault" for sufficiently large $n$. ∎

## B    Formal definition for language $L$

Let $L$ be a language (an extension of a basic formal logical language, $L_0$).

**Definition and base notations.**    The set of $L-$terms is the smallest set $L_t$ such that contain all constant symbols of $L$, all variables, and if $t_1, t_2, ..., t_n$ are in $L_t$ then for any n-ary function symbol $f$, $f(t_1, t_2, ..., t_n)$ is also in $L_t$. Set $L_a$ of atomic formulas are represented by the properties:
(1) if $t_1$ and $t_2$ are terms then $t_1 = t_2$ is in $L_a$, and
(2) the corresponding n-ary function symbols are also in $L_a$.

In other words, the set of all formulas in $L$ (expressions, sentences - herein, we use these interchangeably) is the smallest set containing all atomic formulas and closed under logical connectives $\vee, \wedge, \neg, \rightarrow, \leftrightarrow$, quantifiers $\exists, \forall$, equality symbol " $=$ ", parenthesis "(" and ")", and variables. For our purposes herein and simplicity, it is sufficient to consider that theory in language $L$ is a set of sentences in first-order logic over $L$. We also assume first-order logic with equality; in other words, only normal models are employed. Thus, the models, considered herein (e.g., Erdős–Rényi or finite graph model for random graphs, are normal).

The main language in this paper is that of graphs [4] . We denote $\mathbb{G}_L$ the first-order theory over language of graphs $L$. One convenient (and usual) laxity talking about expressions and formulas in $L$ is using $L$ and $\mathbb{G}_L$ interchangeably.

## C    0-1 law for graphs $L$

We introduce a few known formulations for the 0-1 law for finite graphs.

**Lemma C.1.** *0-1 Law.    For any first-order formula $\phi$ and graph $G$ in $\mathbb{G}_L$ (with the equivalent notation $\mathcal{G}(\omega)$ which is intuitively more suitable), let*

$$G_{n,\phi} = \frac{|\{G \models \phi : |G| = n \text{ and } G \text{ is a graph}\}|}{|\{G : |G| = n \text{ and } G \text{ is a graph}\}|} \tag{9}$$

*Then $\lim\limits_{n \to \infty} G_{n,\phi}$ is 0 or 1.*

*Proof.*  Refer to, e.g., (Fagin, 1976). ■

This can be reformulated as

**Lemma C.2.** *0-1 Lemma.    For any property $\mathcal{A}$ that can be described by a first-order expression $\phi$ and $G_n = \{G : |G| = n$  and $G$ is a graph\},*

$$\lim_{n \to \infty} \mathbb{P}(G_n \in \mathcal{A}) \in \{0, 1\} \tag{10}$$

■

To wit (assuming notations for $G(\omega)$, a set of all finite graphs, and its associated domain $\mathcal{G}(\omega)$, up to isomorphism):

**Lemma C.3.** *Lemma 0.  For any graph $G_n \in \mathcal{G}(\omega)$, $\lim\limits_{n \to \infty} \mathbb{P}(G_n) = 0$ or 1.* ■

The equivalent statement is as follows:  for any first-order expression $\phi$ in theory of $\mathbb{G}_L$, $\lim\limits_{n \to \infty} \mathbb{P}(G_n \models \phi) = 0$ or 1. We can also say that $\lim\limits_{n \to \infty} \mathbb{P}(G_n) \models \phi) = 1 \Leftrightarrow \mathcal{G}(\omega) \models \phi$.

For any *random* graph $G_n \in \mathcal{G}(\omega)$, $\lim\limits_{n \to \infty} \mathbb{P}(G_n) = 0$ or 1.

The equivalent statement is as follows: $\forall$ 0-1 probability $p$ and a first-order expression in theory of graphs, $\phi$, $\lim\limits_{n \to \infty} \mathbb{P}(G_n \models \phi) = 0$ or 1.

---

[4]i.e. graph is a pair $G = (G, E)$ for non-empty set $G$ of nodes (vertices) and a binary relation $E$ on $G$ (the edges). For our purposes, we can assume that $G$ is symmetric and unordered: $E(a, b) \rightarrow E(b, a)$, and $E(a, a)$ is false. We denote $\mathcal{G}(\omega)$ the class of finite graphs and, loosely, the associated first-order logic model, described in the following section C.1 .

We can also say that $\lim_{n\to\infty} \mathbb{P}(G_n \models \phi) = 1 \Leftrightarrow \mathcal{G}(\omega) \models \phi$. One useful representation for the same results is as follows. Given a first-order property $\mathcal{A}$ of a graph $G_n$, $\lim_{n\to\infty} \mathbb{P}(G_n \in \mathcal{A}) \in \{0, 1\}$. Equivalent notation will be $G(n, \omega)$ or just $G(n)$ when the context is clear.

## D    PROOF OF THE MAIN THEOREM

**Theorem D.1.** *No Free Lunch Limitation (NFLL). For almost all proofs, any learning algorithm of inference using randomization in $\mathcal{G}(\omega)$ is almost surely literal learning.*

*Proof.* More instructive than informal considerations is the following proof in which we partially follow a version of the 0-1 law in (Blass et al., 1998). The probability space for the GPT algorithm can be viewed as follows.

**Definition D.2.** Space $\Psi$.

We can assume that any proof (graph labeled by $[n]$) represented in $\mathcal{G}(\omega)$ is encoded into a string. Moreover, due to canonical encoding, we can leverage adjacency information and reduce the context to binary strings. Since we want to include all proofs for a given statement varying by $n$ to be represented (a countable set), we can use a self-delimiting encoding of $n$ followed by adjacency bits as above. That way, every proof is represented by a unique binary string.

Consider a probability distribution over infinite binary strings. Let $\Psi$ be a set of infinite sequences representing proofs (since any string can be encoded by a binary string, in a suitable enumeration (or embedding), and, given a proposition, its proofs of any length can be encoded into an infinite binary string as shown in the previous paragraph).

Let $\Psi$ be a set of infinite sequences $\phi = \langle \phi_n : n \geq 1 \rangle \in \Psi$. In this context, we can view the set as one of independent trials. The resulting probability distribution over $\Psi$ is naturally equipped with the product measure (cf. (Feller, 1968) and Appendix E).

Since we are still in a first-order logic for graphs, $\mathcal{G}(\omega)$, we can consider every proof over strings semantically. Therefore, for any generative algorithm $\mathfrak{A}$, if, given an inference sequence $\phi_n = \{e_0 \to e_1 \ldots e_k \to e_t\}$, representing the proof $\{e_0 \to e_t\}$, we have $\mathfrak{A}(\phi_n) = e_n$, we say that the algorithm succeeds proving $\phi_n$; otherwise, we say it fails.

The corresponding notation for any $\phi \in \Phi$, if $\mathfrak{A}$ succeeds, is $\mathfrak{A} \models \phi$; if $\mathfrak{A}$ fails, we write $\mathfrak{A} \not\models \phi$.

Let us introduce the notation: $p_n(\mathfrak{A}) = \mathbb{P}(\mathfrak{A}$ fails on the n-th step $\phi_n$ of $\phi)$ or $\mathbb{P}(\mathfrak{A} \not\models \phi_n)$ where $\phi$ ranges over $\Psi$.

The following two cases are possible (this is reminiscent of the Borel-Cantelli lemma proof):

**Case 1**. There exists an algorithm, $\mathfrak{A}$ s.t. $\sum_{n=0}^{\infty} p_n(\mathfrak{A}) < \infty$. By the (first) Borel-Cantelli lemma (Feller, 1968), $\mathbb{P}($there are infinitely many $n$ s.t. $\mathfrak{A}$ fails on $\phi_n) = 0$. Thus, for almost all $\phi \in \Psi$, $\mathfrak{A}$ succeeds on all but finitely many $\phi_n$. Therefore, for almost all $\phi$, there exists an algorithm $\mathfrak{A}' = \mathfrak{A} +$ finite lookup that succeeds on $\phi$. The algorithm $\mathfrak{A}$ stays the same for all $\phi$ and only the finite lookup depends on $\phi_n$. It means that, for almost all sequences $\phi \in \Psi$,

$$\mathbb{P}(\mathfrak{A} \models \phi) = 1. \tag{11}$$

The question becomes whether such an algorithm $\mathfrak{A}$ can be GPT. We will show below that the assumption it is GPT meets a contradiction. From (11) we have:

$$\forall \epsilon > 0 \; \exists \; witness \; n_0 > 0 \; s.t. \; \forall n > n_0 \; \mathbb{P}(\mathfrak{A} \models \phi_n) > 1 - \epsilon. \tag{12}$$

On the other hand, from the Accumulating Errors Lemma inequality (2), we see that

$\mathbb{P}(\mathfrak{A} \not\models \phi) > 1 - exp(-\rho)$ where $\rho = \mathbb{P}(\mathbb{E}(\#faults))$.

Thus, setting $\epsilon = 1 - exp(-\rho)$ leads to contradiction with (12). This leaves only two possibilities for the algorithm $\mathfrak{A}$ to generate (since we don't have $\mathbb{P}(\mathfrak{A} \models \phi) = 1$ for almost all $\phi$).

In the first instance, $\mathfrak{A}$ may arrive at nodes representing the false statements, but the inferences would be true (vacuous truths or hallucinations). The proof is still invalid, overall. The second instance is

literal learning; the algorithm may generate a known proof discoverable in the training data for some $\phi$ (potentially, piece-by-piece).

However, in the latter case, given a long enough proof for a novel statement, almost surely, it must encounter a fault.

**Case 2**. For every algorithm $\mathfrak{A}$, $\sum_{n=0}^{\infty} p_n(\mathfrak{A}) = \infty$. Again, as in (2), we can assume that $\phi_n$ are independent events. By the (second) Borel-Cantelli lemma (e.g., (Feller, 1968)), the probability that there exists an infinite number of $n$ that $\mathfrak{A}$ fails on $\phi_n$ is 1. Hence, for every $\mathfrak{A}$ there exists $n$ s.t. $\mathbb{P}(\mathfrak{A} \models \phi_n) = 0$. Since there are only countably many algorithms, for almost all $\phi \in \Phi$, we have:

$$\mathbb{P}(\exists \mathfrak{A}, \ \mathfrak{A} \models \phi) = 0. \tag{13}$$

Qualitatively, this means that in this case, almost surely, no algorithm using randomization with exponential correctness decay can succeed in generating a valid proof for the statement. ∎

## E   NOTATION AND TERMS

GPT/LLM stands for the algorithmic representation of a transformer with attention viewed as the primary inference mechanism for $LLM$.

$\omega$ stands for the first countable ordinal; $[n]$, $\mathbb{N}$ denote a set of natural numbers.

$\mathcal{G}(\omega)$ stands for the infinite set of finite graphs, and the first-order logic model on the set as a domain, where the connectivity of nodes is written as $n_1 \sim n_2$, representing a valid implication $n_1 \implies n_2$.

Formal definitions for the language of the first-order theory can be found in Appendix B. It contains all relevant information on the 0-1 law.

In this setting, a fault in $\mathcal{G}(\omega)$ is an erroneous proof – that is, a chain of thought containing a false implication ($n_1 \nsim n_2$), or a false assumption (node $n_1$ represents falsehood).

**Probability space** $\Psi$ herein is one with the domain of infinite binary strings endowed with product measure homeomorphic to the Borel Probability Space. It is the probability space $([0, 1], \mathfrak{B}([0, 1]), \lambda)$ where $\mathfrak{B}([0, 1])$ is Borel $\sigma$−algebra generated by open intervals and $\lambda$ is a Lebesgue measure.

**"Randomization"** on the probability space $\Psi$ means that the inference mechanism for events admits variability in the token-selection process (e.g., random-seed initialization, temperature, beam search). More formally, the mechanism is a randomized algorithm, i.e., a probabilistic Turing machine $T$ that sends the input string $x$ to the output $T(x)$, which is a random variable. For most purposes in this paper, we do not need to define the precise nature of $GPT$ as an algorithm; it is sufficient to note that it employs (non-deterministic) randomization as described above. Even if the algorithm uses just a pseudorandom seeding, which is a deterministic algorithm, this is not restrictive for our definition. The formalism of $\Psi$ is fully applicable to the context when one operates on the domain of random (finite) graphs, $\mathcal{G}(\omega)$.

A **"Learning algorithm"** is a machine learning algorithm of inference that, after being trained on data, produces output based on the training; for most purposes, an intuitive notion suffices. We consider only learning algorithms that incorporate randomization and have independent, nonzero probabilities of fault at each step.

**"Literal learning"** stands for one which is either memorizes the inferences from a corpus of training/synthetic data (i.e., the memorized or formalized proofs), or vacuous, i.e., $\forall x \ [P(x) \implies Q(x)]$ where $P(x)$ is false for every $x$ or creates a random inference from a false assumption (hallucination), or otherwise invalid.

First-order logic terms can be found in Appendix B.

0-1 Law for graphs is in appendix C.1.

## F  Classifier with undecidable loss

**Theorem F.3.** *Given a domain $D$ of random graphs, the class of binary classifiers over $D$ is (first-order logic) undecidable.* [5]  ∎

## G  Deep Energy-based models (EBMs)

Our main result generally applies to $EBMs$ with randomization. Most contemporary models do use it, since they use stochastic MCMC for sampling. That includes Classical Boltzmann Machine and RBMs. Many use stochastic noise, etc.

However, not all energy-based models use randomization; for example, those employing deterministic gradient flows.

In general, it is a promising direction, since for $EBMs$, inference is optimization-based searching for a configuration with low energy using gradient descent, iterative refinement, and constrained optimization. At the same time, in a reference architecture, $LLMs$ serve as an interface layer for natural-language I/O rather than as a reasoning core. The approach may allow for reducing, and, theoretically, preventing, the accumulation of the error phenomenon that $LLMs$ suffer from.

Recently, startup "Logical Intelligence" announced an excellent result on auto-proving Putnam competition problems (their Aleph agent, paired with GPT-5.2, automatically generated Lean proofs for 668 out of 672 PutnamBench problems (99.4%) in 2026). The post describes Aleph as an orchestration layer and Kona as their proprietary non-autoregressive energy-based reasoning model.

At the time of writing, it is not clear what the exact type of $EBM$ is being used.

## Acknowledgments

We are deeply thankful to anonymous reviewers for insightful notes and thoughtful critiques, which improved the paper.

---

[5](Informally, there is no efficient algorithm that decides whether a well-formed formula in the first-order logic theory of binary classifiers is true. Therefore, if we try to find an "interpretable" explanation for a phenomenon being explained by a model, then, in general, that explanation may not be possible to formulate in first-order logic terms. One may argue that this reduces the very notion of interpretability to heuristics.

