# OpenReview forum: "No Free Lunch GPT/LLM Limitations"
_mathai.club/MathAI/2026/Conference — 2026 Oral_

### Official Review · Reviewer_YcXx · 2026-03-10
**Review: No Free Lunch GPT/LLM Limitations**

**Rating:** 6
**Confidence:** 2

**Review:**

# Rationale:

In this paper, the authors discuss the fundamental limitations of LLMs in constructing correct, long-form mathematical proofs. The main result (Theorem 2.1) leverages the zero-one law within a graph-theoretic model of inference to argue that for sufficiently long or complex proofs, any LLM utilizing randomization will almost surely fail and degenerate into "literal learning" (i.e., memorization or vacuous truth). The paper is well-structured and supported by a range of contemporary examples and benchmarks (e.g., FrontierMath, Aristotle), allowing a non-expert reader to grasp the core problem. The topic is highly relevant to the goals of the conference.

# Remarks Requiring Correction (Minor Revisions):

* Defining "Literal Learning" Earlier: The concept of "literal learning" is central to the entire paper (mentioned early in the Abstract and Intro), but its formal definition is relegated to Appendix E (lines 740-743). Please consider adding a brief, formal definition of this term directly in the main text (e.g., Section 2.1) so readers do not have to flip to the appendix to understand the core claim.

* Bibliographic Inconsistency: There is an error in reference [Asher et al., 2023] (lines 310-312). The provided arXiv ID (2306.12213) links to a paper titled "Limits for learning with Language Models," whereas the bibliography lists "Autocorrelations decay in texts..." as the title. The authors should verify and correct this entry.

P.S. I'm not an expert in this field, so my assessment is based on clarity of presentation, relevance to the conference topic, and the presence of obvious, easily correctable flaws.

---

> ### Author Rebuttal · Authors · 2026-03-13
>
> Thank you very much for your revision suggestions! These are incorporated into revision 2.

---

### Official Review · Reviewer_Nisj · 2026-03-13
**The paper argues that modern transformer-based language models (LLMs/GPT) are inherently incapable of reliably generating complex mathematical proofs due to architectural limitations related to stochastic inference and token-level prediction. The authors formalize reasoning as graph-based inference and attempt to prove a “No Free Lunch”–style theorem stating that any randomized inference algorithm in such a framework almost surely reduces to literal learning (memorization) when producing valid proofs. While the paper raises an important and timely question regarding the limits of LLM reasoning, its theoretical argumentation is often informal, relies on strong assumptions, and lacks rigorous justification for mapping LLM inference to the proposed graph and probability framework. Consequently, the central theorem is not convincingly established. The paper’s conceptual discussion is interesting, but the mathematical claims and empirical support are insufficient for strong acceptance.**

**Rating:** 6
**Confidence:** 4

**Review:**

**Brief Summary**

The paper argues that modern transformer-based language models (LLMs/GPT) are inherently incapable of reliably generating complex mathematical proofs due to architectural limitations related to stochastic inference and token-level prediction. The authors formalize reasoning as graph-based inference and attempt to prove a “No Free Lunch”–style theorem stating that any randomized inference algorithm in such a framework almost surely reduces to literal learning (memorization) when producing valid proofs. While the paper raises an important and timely question regarding the limits of LLM reasoning, its theoretical argumentation is often informal, relies on strong assumptions, and lacks rigorous justification for mapping LLM inference to the proposed graph and probability framework. Consequently, the central theorem is not convincingly established. The paper’s conceptual discussion is interesting, but the mathematical claims and empirical support are insufficient for strong acceptance.

**Detailed Review**

**Overview**

The paper addresses the limitations of large language models in rigorous reasoning and mathematical proof generation. It attempts to demonstrate that these limitations are not merely empirical shortcomings but rather inherent consequences of the GPT/LLM architecture.

The core idea is to model inference as paths in a finite graph (G(\omega)) where nodes represent statements and edges represent logical implications. The authors use results related to zero-one laws for random graphs and probabilistic arguments about error accumulation to claim that:

- Any randomized inference algorithm operating in such a setting will almost surely fail for sufficiently complex proofs.
- The only scenario where valid proofs are produced is “literal learning” -- essentially memorization or reproduction of training data.
- Therefore, transformer-based architectures are fundamentally unsuitable for discovering novel mathematical results.

The paper presents several theoretical results, including:

- Theorem 2.1 / Theorem D.1: Randomized inference algorithms almost surely reduce to literal learning.
- Accumulating Errors Lemma: The probability of error-free reasoning decays exponentially with proof length.
- Additional arguments referencing Gödel incompleteness, zero-one laws, and random graph theory.

The paper concludes that LLMs can still serve as assistive tools but are unlikely to autonomously generate novel mathematical discoveries of significant complexity.

**Strengths**

1. The paper addresses a highly relevant problem: whether LLM architectures can fundamentally support reliable logical reasoning and proof generation. This question is central to current AI research and mathematical AI.
2. Unlike many empirical papers on LLM reasoning, this work attempts to provide a formal theoretical argument rather than relying purely on benchmarks. The use of graph representations of proofs, probabilistic reasoning, references to zero-one laws, and discussion of Gödel incompleteness shows ambition in grounding the argument in mathematical logic.
3. The paper clearly articulates its central thesis "stochastic next-token inference leads to error accumulation and therefore prevents reliable generation of long proofs". This idea is intuitively plausible and aligns with several empirical observations about LLM reasoning failures.
4. The paper makes a reasonable critique of current benchmarks (e.g., mathematical benchmarks becoming saturated or contaminated by training data). This is an important issue in evaluating reasoning models.

**Weaknesses**

Despite the interesting premise, the paper has several major theoretical and methodological weaknesses.

1. The mapping between LLM inference and the graph model is asserted but not formally established. Modeling proofs as simple graph paths ignores hierarchical proof structures, symbolic reasoning modules, external tools (retrieval, theorem provers), and verification loops. Thus this mapping does not capture the real capabilities of modern AI systems.
2. The argument relies heavily on informal reasoning and assumptions about independence of errors, which are unrealistic in neural networks. In real models errors are correlated, attention mechanisms allow cross-step context, and reasoning processes may self-correct.
3. The application of zero-one laws for random graphs to LLM inference is not rigorously justified.
4. The result appears closer to a heuristic argument than a formal theorem.
5. The paper claims that "post-training methods, RLHF, MoE, retrieval, etc., cannot overcome these limitations. However, the theoretical analysis does not formally include these architectures. The conclusions therefore appear overstated. In particular, the analysis does not consider tool-augmented systems, formal verification loops, symbolic reasoning modules, and planning architectures.
6. The paper provides no experiments or quantitative validation of its claims. The threshold estimates (e.g., 18k lines of Lean code or 27 pages of natural language) are described as coming from “experimentation” but methodology is not described, datasets are unspecified, and results are not reproducible. Given the strength of the theoretical claims, empirical evidence would be necessary.
7. Several classical results (zero-one laws for random graphs, Borel–Cantelli lemma, Gödel incompleteness theorem) are invoked but their applicability is unclear. The connection between these results and transformer inference is mostly analogical rather than formal. In particular, Gödel incompleteness does not imply that stochastic reasoning systems cannot produce valid proofs, and zero-one laws describe asymptotic graph properties, not learning systems.
8. The paper does not sufficiently discuss related work on neuro-symbolic reasoning, formal proof assistants integrated with LLMs, program synthesis approaches, and verifier-guided reasoning systems. These systems directly address the limitations the paper claims are fundamental.

**Suggestions for Improvement**

1. The core theoretical framework needs a precise mapping from transformer inference to the proposed graph structure.
2. The error accumulation model should account for correlated reasoning steps and self-correction mechanisms.
3. The paper should specify whether the theorem applies only to autoregressive transformers, purely stochastic inference systems, and systems without verification or external tools.
4. The claims about proof-length thresholds and exponential failure should be validated with controlled experiments.
5. The paper should discuss discuss related work on neuro-symbolic systems, theorem-proving pipelines, and tool-augmented LLM reasoning.
6. If the results are intended as conceptual arguments, they should be presented as conjectures rather than rigorous proofs.

---

> ### Author Rebuttal · Authors · 2026-03-13
>
> Thank you very much for your thoughtful and thorough review! We certainly appreciate the suggestions, some of which are being incorporated into revision 2 as we speak. Our full (detailed) rebuttal will follow shortly below.

---

### Author Rebuttal · Authors · 2026-03-13

$Introduction$

The paper clearly establishes the boundaries between informal considerations (designed to develop a reader's intuition) and formal results. It is mainly written for the Computer Science audience. That is a justification of the style and a manner of argumentation.
From that perspective, the paper clearly says that the reader can immediately refer to the rigorous proof of the main result. Point $7$ in the reviewer's $"Weaknesses"$ section is surprising. Borel-Cantelli theorems (or lemmas, in some reformulations, if preferred) are the instrumental part of the proof. Zero-one laws are part of the first-order logic representation of inference. Gödel's incompleteness theorem is used to prove an undecidability result, illustrating why, for some problems, planning and scaffolding are necessary for successful automated inference.

The reviewer then states, "The connection between these results and transformer inference is mostly analogical rather than formal". Isn't the statement like this informal since it requires a formal proof? The whole point of the paper is that such a connection exists.

The section "Detailed Overview" fairly presents the paper's main motives and results.
Yet the core idea is misrepresented. The idea is to use the Borel Probability space and an analog of the Borel-Cantelli argument for the $0-1$ law to link the accumulation of errors with the inherent limitation in question. Random graphs are simply one of the tools to bring about a more intuitive representation down the road.

$Detail Rebuttal$ to the section $“Strengths”$
N/A

Detail Rebuttal to the section $ "Weaknesses" $  follows the reviewer's numeration.

1.	LLM inference can be represented logically in a first-order logic of finite graphs by choosing an appropriate level of abstraction. The argument that the model ignores high-level logic constructs and structures is therefore irrelevant. The argument that $G(\omega)$ "does not capture capabilities of modern AI systems" is like saying that a Turing machine does not capture computation. One can ignore Church's thesis, but that would not be practical.

2.	There is enough argumentation in the paper why independence of errors can be assumed, given that the attention window is finite and almost surely a type of theorem. The argument of self-correction is beside the point. Self-correction doesn't eliminate all errors for the very statement of the main theorem.

3.	Argument no 1 above suffices.

$To \ be \ continued.$

---

> ### Author Rebuttal · Authors · 2026-03-13
>
> $Cont.$
>
> 4.	Valid observation, although the paper made the demarcation between the informal and formal explicit.
> 5.	One doesn't need to jump into the particulars of the model inference recovery. The beauty of 1st-order logic is that there is no need to bring them in to trace certain phenomena with respect to correctness. Why? Because those use randomization and are error-prone.
> 6.	This is a valid statement since the results are coming in a separate paper. It is an oversight. It makes sense to remove them from the paper altogether. Bringing details, etc. It will make the paper too large for its purpose. We were just eager to point out when verifiable, rigorous, autonomous GPT/LLM inference likely becomes prohibitive. Thank you for this useful suggestion.Removing in version 2.
> 7.	 Addressed above
> 8.	These areas are "nice to have, so to speak. But the same rebuttal as in 6 applies.
>
> $Rebuttal$ to the section: $"Suggestions for Improvement."$ follows the reviewer's numeration.
>
> 1.	 We addressed it above. The graph formulation is advantageous, but not necessary for a rigorous proof of the main result.
> 2.	None of these refute the premises of the accumulation Lemma for a simple reason. These mechanisms are error-prone, and with a large enough window, the premises of the accumulation Lemma remain true. Therefore, the adopted model suffices.
> 3.	The paper refers to GTP/LLM, so an autoregressive architecture is assumed. One has to be careful to reformulate the theorem in the present state for stochastic inference systems, since they can use probabilistic models without randomization. It is clear from the exposition of the paper under discussion that we assume autonomous inference, which is why we argue that the main application of GPT/LLM reasoning is to assist an expert. However, it is a valid and useful suggestion, and we intend to incorporate this into the paper. Thank you for the suggestion.
> 4.	Removed from version 2, following the suggestion.
> 5.	 Addressed above.
> 6.	Hopefully, it is clear from the considerations above that we intend the main result to be a $theorem$ rather than a $conjecture$.
>
> $Appendix$
> Suggestion for improvement no. 5 is useful and could be a subject for a separate paper that presents neuro-symbolic reasoning, LLM-based theorem proving, and tool-augmented inference. The closest match appears to be https://arxiv.org/html/2505.14479v1, but this paper does not have formal results. In formal settings, it appears novel.

---

### Decision · Program_Chairs · 2026-03-14

**Decision:**

Accept (Oral)

**Comment:**

Dear Author(s),

On behalf of the Program Committee of the International Conference on Mathematics of Artificial Intelligence (MathAI 2026), we are pleased to inform you that your paper has been accepted for an oral presentation at MathAI 2026.

Your paper was evaluated through a rigorous two-stage review process involving both automated screening and expert review by members of the Program Committee. The reviewers recognized the quality and contribution of your work.

Presentation details:

- Format: Oral presentation (15–20 minutes + 5 minutes Q&A)
- Mode: You may present either in person (offline) at the conference venue in Sirius, Russia, or remotely via Zoom. Please indicate your preferred mode when confirming your participation.
- Conference dates: Marh 30 - April 3, 2026
- Website: https://mathai.club

Next steps:

1. Please confirm your participation and presentation mode by replying to this email mathai.club@yandex.ru no later than March 15, 2026 18:00 Moscow time.
2. If you plan to attend in person, the organizing committee will provide accommodation details separately.
3. Please prepare your final camera-ready manuscript according to the formatting guidelines available at https://mathai.club and upload it to OpenReview by March 15, 2026 18:00 Moscow time.

Should you have any questions regarding the program, logistics, or your presentation slot, please do not hesitate to contact us.

We look forward to your contribution to MathAI 2026.

With kind regards,

MathAI 2026 Program Committee
International Conference on Mathematics of Artificial Intelligence
https://mathai.club
OpenReview: https://openreview.net/group?id=mathai.club/MathAI/2026/Conference
Telegram: https://t.me/MathAI_club
Email: mathai.club@yandex.ru